# Flow Boiling Heat Transfer Intensification Due to Inner Surface Modification in Circular Mini-Channel

Aleksandr V. Belyaev [1,*], Alexey V. Dedov [1], Nikita E. Sidel'nikov [1], Peixue Jiang [2], Aleksander N. Varava [1] and Ruina Xu [2]

[1] Department of General Physics and Nuclear Fusion, The National Research University "MPEI", Krasnokazarmennaya 14, 111250 Moscow, Russia
[2] Department of Thermal Engineering, Tsinghua University, Haidian District, Beijing 100084, China
* Correspondence: beliayevavl@mpei.ru

**Abstract:** This work aimed to study the intensification of flow boiling heat transfer and critical heat flux (CHF) under conditions of highly reduced pressures due to a modification of the inner wall surface of a mini-channel. Such research is relevant to the growing need of high-tech industries in the development of compact and energy-efficient heat exchange devices. We present experimental results of the surface modification effect on hydrodynamics and flow boiling heat transfer, including data on the CHF. A description of the experimental stand and method for modifying the test mini-channel is also presented. The studies were carried out with freon R-125 in a vertical mini-channel with a diameter of 1.1 mm and a length of 50 mm, in the range of mass flow rates from $G$ = 200 to 1400 kg/(m²s) and reduced pressures between $p_r = p/p_{cr}$ = 0.43 and 0.56. The maximum surface modification effect was achieved at a reduced pressure of $p_r$ = 0.43, the heat transfer coefficient increased up to 110%, and the CHF increased up to 22%.

**Keywords:** flow boiling; critical heat flux; heat transfer enhancement; high reduced pressure; mini-channel





## 1. Introduction

One way to increase the heat flux density during flow boiling is surface modification. The increased interest in studying the methods that intensify heat transfer (primarily due to boiling) is associated with the increasing requirements for overall efficiency and reduced environmental impact in modern energy-efficient and energy-intensive installations. The requirements for heat removal in microelectronics elements increase the heat flux density, which reaches 2–5 MW/m². To provide the necessary heat removal in such devices, highly efficient mini-channel heat exchangers are being developed and implemented [1,2], where various dielectric liquids and freons can be used as coolants. It is also possible to use boiling as an efficient mechanism for heat removal and additional intensification by modifying the boiling surface.

*Surface Modification Methods*

Technological growth in areas associated with high heat flux density requires the creation of appropriate heat exchangers. To date, there are many articles that describe various methods for modifying the boiling surface [3]. The main part of the intensification methods was laid down in the twentieth century. However, the development of surface treatment technologies and methods for measuring the characteristics of non-stationary processes, as well as simulation techniques, allows modern research to be carried out at a fundamentally new level. Improved methods of mechanical surface treatment can achieve a significant increase in heat flux. One of the long-standing and well-known ways to increase the heat flux during fluid flow in a channel is by changing the relief using the method of deforming cutting (MDC) [4–6]. This method of obtaining modified surfaces is patented and is actively used in the commercial sphere and in experimental studies. For example, in

the article [7], experiments were carried out on low-finned pipes and pipes of the Turbo-B and Thermoexcel-E types, with a length of 152 mm and an outer diameter of 18–19 mm, under conditions of free convection at a saturation temperature of 7 °C for the refrigerants HCFC22, HFC134a, HFC125 and HFC32. Heat fluxes varied in the range of 10–80 kW/m². The ranges of increase in the heat transfer coefficient during boiling on a low-finned pipe, the Turbo-B type, and the Thermoexcel-E type were 1.09–1.68; 1.77–5.41 and 1.64–8.77 times, respectively. A combined modification (simultaneous use of MDC and nanocoating) was performed in [8]. The heat exchange surface was made in the form of channels, onto which a nanocoating was applied. The channel depth and rib width for all surfaces were kept constant at 400 μm and 200 μm, respectively. In comparison with a smooth surface, the increase in CHF was 228%.

Experimental investigations of surface modifications are carried out in pool boiling. There are many works devoted to various methods of intensification [9]. Modern methods of plasma and ion sputtering, chemical processing technologies, as well as various methods for creating nanoscale coatings and their combination with microstructures, make it possible to create fundamentally new types of multiscale structured surfaces. There are many different methods for surface modification: meshed surfaces [10,11], silicon oxide nanoparticle coated copper [12], cutting with a micro milling machine [13], and the use of micro-nanostructured surfaces [14–17] (which make it possible to achieve an increase in the heat transfer coefficient (HTC) of up to 200% and in the CHF of up to 120%), the use of capillary-porous structured surfaces [18] (HTC increases from 1.5 to 4 times). There are methods of cold deposition by McNamara [19], including the creation of carbon [20–22] and silicon nanotubes [23], which increase the HTC up to 3 times and the CHF up to 65%. The methods of modification using a laser, for example, laser T-shaped groove [24] and laser surface texturing [25], can increase the CHF up to 240%. Due to a very high HTC, these laser technologies have found application in many industrial fields [26]. Another common method of surface modification is the use of mesh coatings. Pool boiling on a modified surface was studied in [27] using multilayer copper gradient grid coatings. The results indicated an increase in the CHF of 3 times and an increase in the HTC of 6.6 times.

All the above methods of surface modification have been studied for pool boiling. However, it is difficult, or impossible, to implement these methods in channels. Therefore, there are currently significantly fewer methods for surface modification in channels. Grooving technology can be used to modify the inner wall of a circular channel. This technology was studied in [28], where pipes with a diameter of 9.52 mm were used with different surface structures: a smooth copper pipe and two copper pipes with internal grooves of different cut widths. The modification technology creates grooves along the inner wall of the channel, which, as a result, increases the heat exchange surface area. This method is simple to implement and does not cause considerable technical difficulties. However, the modifications have a strong influence on the channel hydrodynamics, where the pressure drop is more than twice as high as in a smooth pipe. According to [28], the best results can be obtained in pipes with the smallest groove width, which showed an increase in the HTC of more than twice compared to smooth pipes. The maximum increase in the HTC was observed at the vapor quality $x = 0.8$.

The flow boiling heat transfer intensification in a rectangular channel measuring 65 mm long, 6 mm wide, and 1 mm high was studied in [29]. One of the walls (through which the flow was heated) was made of silicon, on which a C4F8 polymer layer was deposited using a mask. The result was a bifilar surface. FC-72 was used as the working fluid. The experiments were carried out at two values of the mass flow rate: $G = 90$ and $130 \ kg/m^2 s$. An increase in the HTC of 26% was obtained at heat flux density $q = 9 \ W/sm^2$.

There are many methods of modification for pool boiling which make it possible to create effective intensifying surfaces that significantly increase heat transfer during boiling and evaporation. Currently, there are a small number of methods for surface modification in channels, especially in mini- and micro-channels. The aim of this work is to use a relatively simple method for modifying the inner wall of a mini-channel to intensify flow boiling.

## 2. Experimental Methods and Setup Description

### 2.1. Experimental Setup

An experimental setup was used to study the processes of hydrodynamics and heat transfer in the mini-channel at highly reduced pressures. A detailed description is presented in earlier works on the experimental study of the hydrodynamics, heat transfer, and CHF for a wide range of mass flow rates, vapor qualities, and reduced pressures [30,31], as well as at the most demanded flow parameters in technology [32]. The experiments were carried out using R-125, the properties of which are presented in Table 1. The saturation temperature for the operating parameters of the performed experiments was in the range of 30–45 °C, which made it possible to reduce heat losses to the environment during boiling in the channel. Also, the heat of vaporization and the critical pressure was much lower than in water, which made it possible to achieve the required parameters with less energy.

**Table 1.** Physical properties of R-125.

| | |
|---|---|
| Saturation temperature (1 atm), °C | −48.1 |
| Critical point pressure, MPa | 3.618 |
| Liquid density (25 °C), kg/m$^3$ | 1190 |
| Vapor density (boiling temperature), kg/m$^3$ | 6.7 |
| Surface tension (25 °C), N/m | 0.014 |
| Liquid specific heat, kJ/(kg·K) | 1.399 |
| Latent heat of vaporization (25 °C), kJ/kg | 164 |
| Liquid thermal conductivity (25 °C), W/(m·K) | 0.062 |
| Liquid viscosity (25 °C), mPa·s | 0.141 |

The working tube was placed vertically and made of 12 × 18H10T stainless steel with a heated length of 50 mm, an inner diameter of 1.1 mm, and an outer diameter of 1.6 mm. The tube was heated by alternating the current. The inlet and outlet pressures and pressure drops were measured using pressure sensors and a differential manometer. The inlet and outlet temperatures were measured using Chromel-Copel thermocouples with a cable diameter of 0.7 mm. The wall temperatures were measured using Chromel-Copel thermocouples (diameter 0.2 mm) on five cross-sections of the tube (T1–T5, see Table 2). The inner wall temperatures were calculated using a correction for the wall conductivity. Heat losses were taken into account when calculating the heat balance of regimes with convective heat transfer. An estimate of the measurement uncertainty is presented in Table 3.

**Table 2.** Coordinates of the cross-sections (mm).

| T1 | T2 | T3 | T4 | T5 |
|---|---|---|---|---|
| 2.5 | 15.5 | 28.5 | 40 | 48 |

**Table 3.** Uncertainty parameters in the analysis.

| Parameter | Uncertainty |
|---|---|
| Current | ±0.9% |
| Voltage | ±0.5 mV |
| Mass flow rate | ±0.2% |
| Inlet and outlet temperatures | ±0.1 °C |
| Wall temperature | ±0.8% |
| Inlet and outlet pressure sensors | ±1% |
| Pressure drop sensor | ±0.2% |
| Tube diameter | ±0.05 μm |

### 2.2. Surface Modification

Modification of the inner wall was carried out using the action of a laser pulse on the outer surface of the tube. As a result, exposure formations of different heights and diameters, depending on the pulse power (p1–p14), were formed on the inner wall (Figure 1a). At some values of the pulse power, the outer surface of the tube was not damaged (Figure 1b).

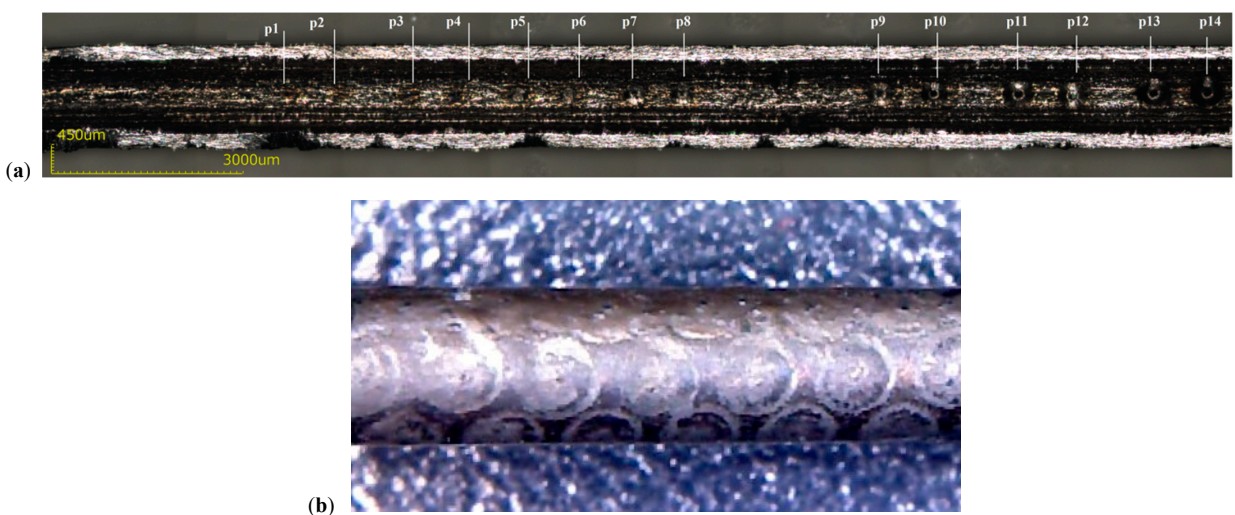

**(a)**

**(b)**

**Figure 1.** (**a**) View of the inner surface of the wall after exposure to laser pulses of different powers. (**b**) External surface after exposure with the technological current value $I$ = 130 A.

Figure 2 shows a photograph and a formation profile of the inner wall of the mini-channel after the outer wall was exposed to a laser pulse with a technological current value of $I$ = 130 A. The diameter of the formation was d = 390 μm. The formations have the shape of a rounded cone with a dip in the center, located at the point of the laser pulse impact. At such a power, pores and craters with different diameters, ranging from 5 to 60 microns, form on the upper part of the cone. The size distribution of the pores is most uniform in the center of the dip where their diameter is about 15 μm.

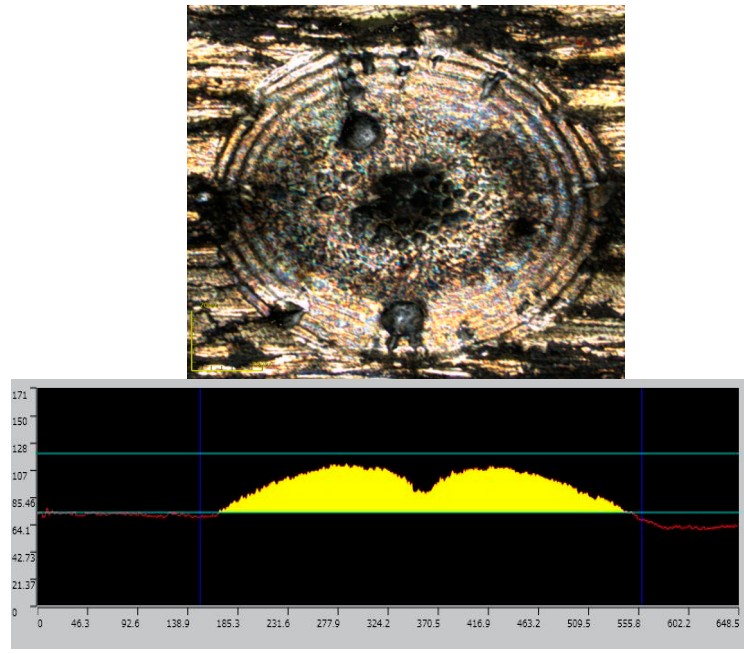

**Figure 2.** Photograph and view of the formation profile along the tube resulting from exposure to a laser pulse of $I$ = 130 A, $d$ = 390 μm.

At the pulse current $I = 110$ A, the formation diameter decreased to 240 μm, and the pattern of the crater location changed. Figure 3 shows that large pores with a diameter from 15 to 30 μm are located both in the peripheral and the central regions. The pores with the smallest diameter from 2 to 10 μm are formed in the annular region between the center and periphery. At current $I = 110$ A, the average pore diameter is smaller than at current $I = 130$ A.

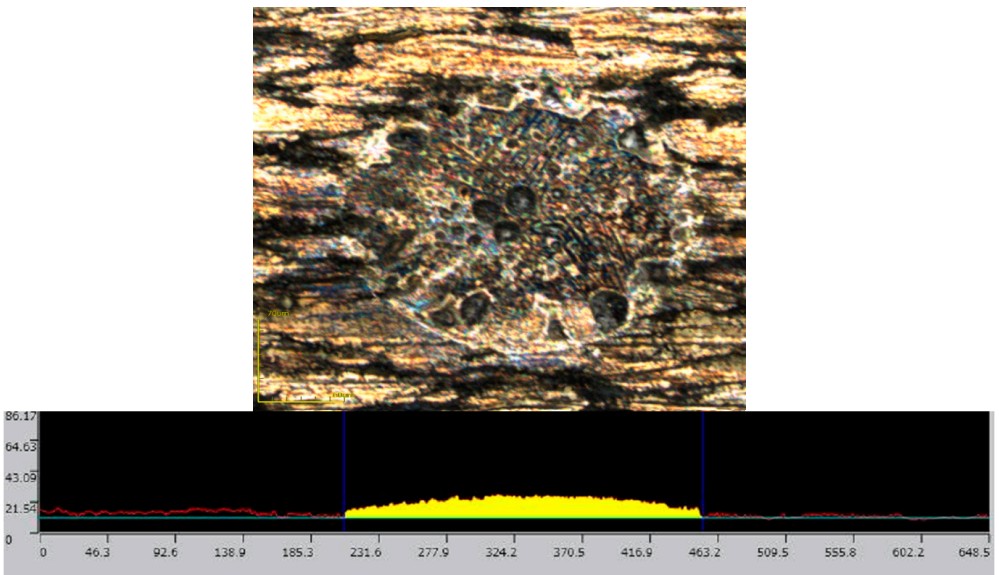

**Figure 3.** Photograph and view of the formation profile along the tube resulting from exposure to a laser pulse of $I = 110$ A, $d = 240$ μm.

The parameters of the laser pulse were selected based on the results of the analysis of the obtained formations. In this work, the channel modification was performed using a laser pulse with the technological current $I = 130$ A. Along the inner wall surface of the mini-channel, the laser pulse made formations with a high density of arrangement in the amount of 300 pieces (6 rows of 50 formations in each, in a checkerboard pattern, see Figure 1b).

## 3. Experimental Results

The experiments were carried out with R-125 in a vertical channel with a diameter of 1.1 and length of 50 mm at two values of reduced pressure: 0.43 ($T_s = 30$ °C) and 0.56 ($T_s = 40$ °C). Regimes of convective and flow boiling heat transfer were obtained in each experiment. The mass flow rate varied in the range of $G = 200–1400$ kg/m$^2$s. The inlet flow temperature was close to room temperature ($T_{in} \approx 25$ °C). The values of inlet and outlet temperatures, the wall temperature at five sections along the tube, the inlet and outlet pressures, the pressure drop, and the mass flow rate were measured. The measurements were obtained using an automated data acquisition system after a stationary regime was established. The maximum heat flux was limited by the critical heat flux. The CHF was observed and recorded as a sharp increase in the wall temperature according to the readings of the thermocouple located near the outlet of the tube.

### 3.1. Effect of the Modification on the Hydrodynamics

The effects of the modification on the channel hydrodynamics were studied without heating the test tube. The total pressure drop, including the inlet and outlet pressure drops, was measured at a constant inlet temperature with an increasing mass flow rate. The

prediction of the total pressure drop was carried out as follows. The pressure drop in the heated part of the mini-channel was calculated using Formula (1):

$$\Delta p_{tube} = \xi \frac{\rho w^2}{2} \frac{L}{d}, \ \xi = (1.82 \lg(Re) - 1.64)^{-2}. \tag{1}$$

The test tube had unheated parts at the inlet and outlet, which passed into supply tubes with a diameter of 4 mm, forming a diffuser and confuser. Formula (1) was used to calculate the pressure drop on the unheated parts. The calculation methods of [33] were used to calculate the pressure drop in the confuser and diffuser. The resulting pressure drops at the inlet and outlet were calculated using Formulas (2) and (3):

$$\Delta p_{inlet} = 0.5 \left(1 - \frac{d^2}{D^2}\right)^{\frac{3}{4}} \frac{\rho w^2}{2} \xi \frac{\rho w^2}{2} \frac{L_{in}}{d}, \tag{2}$$

$$\Delta p_{outlet} = \left(1 - \frac{d^2}{D^2}\right)^{2} \frac{\rho w^2}{2} + \xi \frac{\rho w^2}{2} \frac{L_{out}}{d}, \tag{3}$$

where $L_{in}$ and $L_{out}$ are the lengths of the inlet and outlet unheated parts, respectively; $d$ is the diameter of the mini-channel; and $D$ is the diameter of the supply tubes.

In Figure 4, the filled triangles show the experimental data of the total pressure drop without modification. The line shows the sum of the calculated pressure drops in all three parts according to Formula (4):

$$\Delta p_{calc} = \Delta p_{inlet} + \Delta p_{outlet} + \Delta p_{tube}. \tag{4}$$

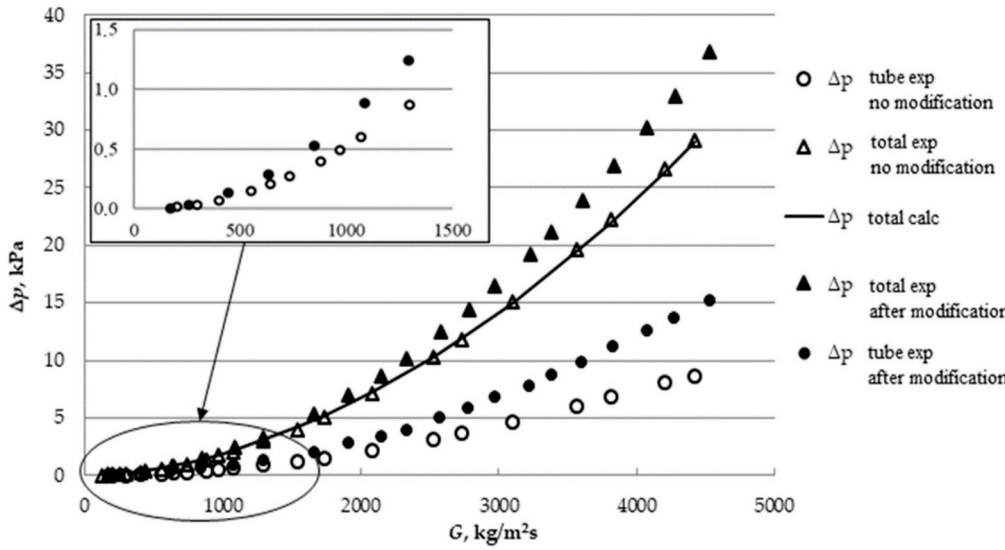

**Figure 4.** Pressure drop versus mass flow rate.

It is clear that the difference between the experimental data and the calculated data is insignificant, which suggests that the calculation of pressure drop at the inlet and outlet of the tube was carried out correctly. Thus, from the experimental data on the total pressure drop obtained before and after the modification, the experimental pressure drops in the heated part of the mini-channel were determined using Formula (5):

$$\Delta p_{tube.exp.} = \Delta p_{exp} - \Delta p_{inlet} - \Delta p_{outlet} \tag{5}$$

In Figure 4, the unfilled dots show the pressure drop in the tube without modification, whereas the black dots show the pressure drop after the modification. The greatest increases in the pressure drop of 50% to 90% were observed at mass flow rates of $G > 2000$ kg/m²s.

The maximum pressure drop did not exceed 40% within the range of mass flow rates that was investigated in the current work (200–1400 kg/m²s).

### 3.2. Heat Transfer Enhancement

Primary data on the temperature at the inlet and outlet of the tube, the wall temperature at five cross sections, and the heat flux density at various mass flow rates and pressures were obtained during the experiments. In each regime, at fixed values of the mass flow rate and the inlet temperature and pressure, the tube was gradually heated until it reached the CHF. Figure 5 shows an example of the wall temperature versus the heat flux density in section T4 before and after modification at $p_r = 0.56$ and $G = 1260$ kg/(m²s). The regimes of convective and boiling heat transfer are clearly visible.

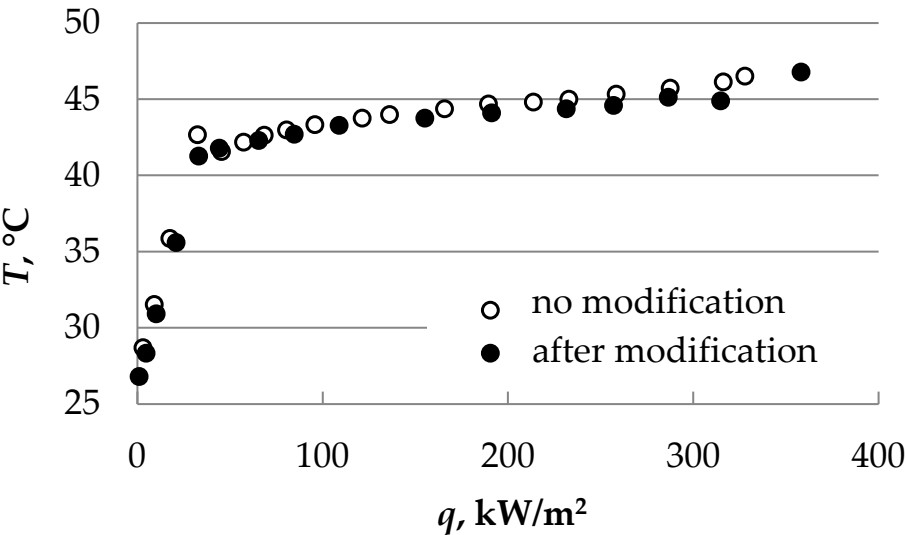

**Figure 5.** Wall temperature in the T4 thermocouple section versus heat flux density.

Based on the obtained data, the heat transfer coefficients for the regimes of convective and boiling heat transfer were calculated using $h_{boil} = q/(T_w - T_l)$ and $h_{con} = q/(T_w - T_s)$, where $T_w$, $T_l$, and $T_s$ are the wall, liquid, and saturation temperatures, respectively. The HTC obtained before the tube modification was then compared to the HTC after modification.

Figures 6–8 show the results of the effect of the modification on the HTC in nucleate boiling regimes for the mass flow rates $G = 420, 800, 1260$ kg/(m²s) obtained at reduced pressures of $p_r = 0.43$ and $p_r = 0.56$. The HTC with increasing heat flux density and mass flow rate increased as usual. A decrease in the HTC was observed upon reaching the pre-crisis values of the heat flux. As a result of the modification to the mini-channel, the nucleate boiling HTC increased. The greatest effect of the modification was observed at $p_r = 0.43$ for the mass flow rate $G = 1260$ kg/(m²s) where the average increase in the HTC was 110% (Figure 8a). The average increase in the HTC for mass flow rates $G = 420$ kg/(m²s) (Figure 6a) and $G = 800$ kg/(m²s) (Figure 7a) were 65% and 100%, respectively.

The results obtained at the reduced pressure $p_r = 0.56$ are similar, but with a lower degree of intensification. The greatest effect of the modification at a highly reduced pressure was observed at $G = 800$ kg/(m²s) where the average increase in the HTC was 23% (Figure 7b). For mass flow rates $G = 420$ and $1260$ kg/(m²s), the average increase in the HTC was 11% and 14%, respectively. With increasing pressure, the critical diameter of vapor bubbles decreases, which leads to an increase in the density of vaporization centers due to natural roughness. This method of surface modification at the highly reduced pressure $p_r = 0.56$ probably becomes less effective because the size of some pores is too large in relation to the critical bubble diameter to be the centers of vaporization.

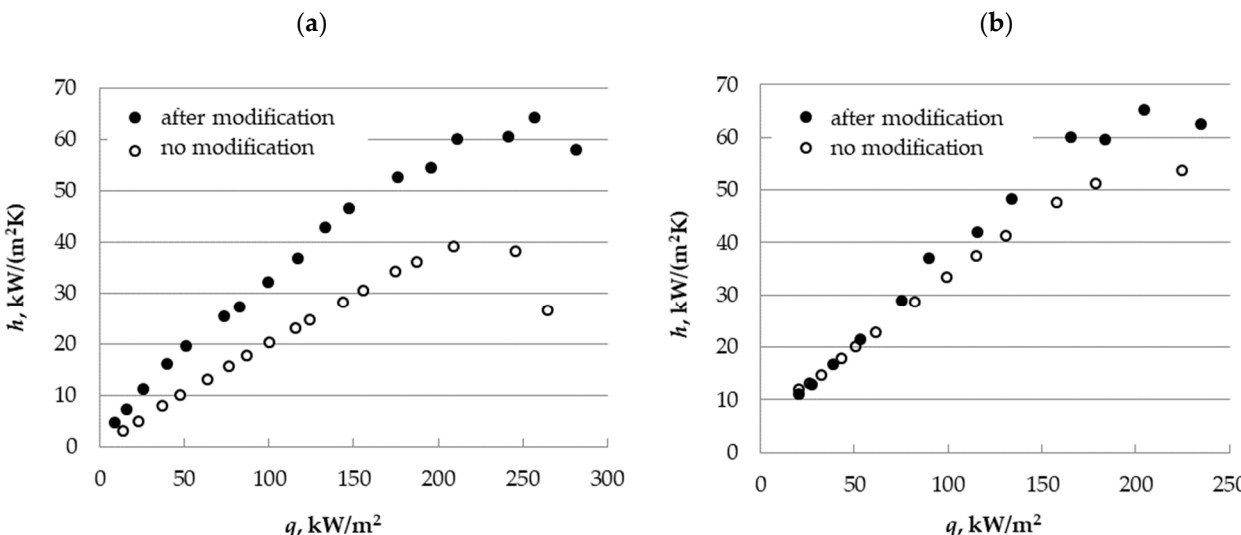

**Figure 6.** Effect of the modification on HTC at $G$ = 420 kg/(m²s) in the T4 thermocouple section, (**a**) $p_r$ = 0.43, (**b**) $p_r$ = 0.56.

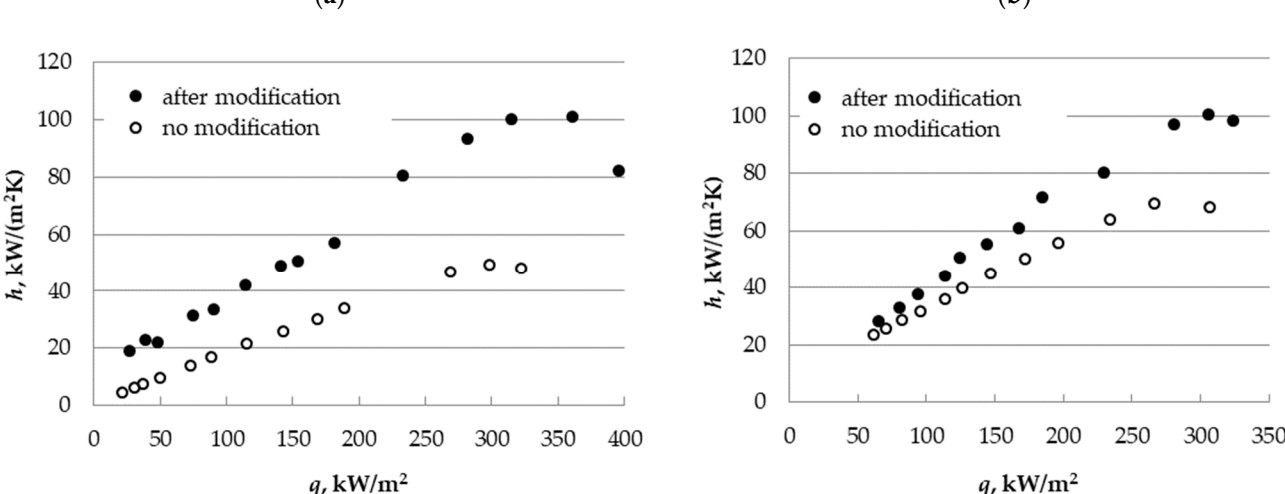

**Figure 7.** Effect of the modification on HTC at $G$ = 800 kg/(m²s) in the T4 thermocouple section, (**a**) $p_r$ = 0.43, (**b**) $p_r$ = 0.56.

The heat transfer data were obtained at different pressures. It is clearly seen from the data analysis that an increase in the reduced pressure from $p_r$ = 0.43 to $p_r$ = 0.56 caused the HTC to increase significantly, on average from 25% to 100%, depending on the mass flow rate (the higher the mass flow rate, the greater the increase in the HTC). The greatest effect of the modification of the inner wall of the mini-channel on the intensity of flow boiling heat transfer was observed in the range of the mass flow rate $G$ = 1260 kg/(m²s) at the reduced pressure $p_r$ = 0.43. The HTC data obtained on the modified tube at the reduced pressure $p_r$ = 0.43 are comparable with the data obtained before the modification at the pressure $p_r$ = 0.56. Increasing the pressure from $p_r$ = 0.43 to $p_r$ = 0.56 led to an increased HTC in the tube without modification, which, on average, was comparable to the increase in the HTC due to surface modification at $p_r$ = 0.43.

To evaluate the effect of the modification on convective heat transfer, the values of the experimental convective HTC obtained before and after the modification at the same flow parameters were compared.

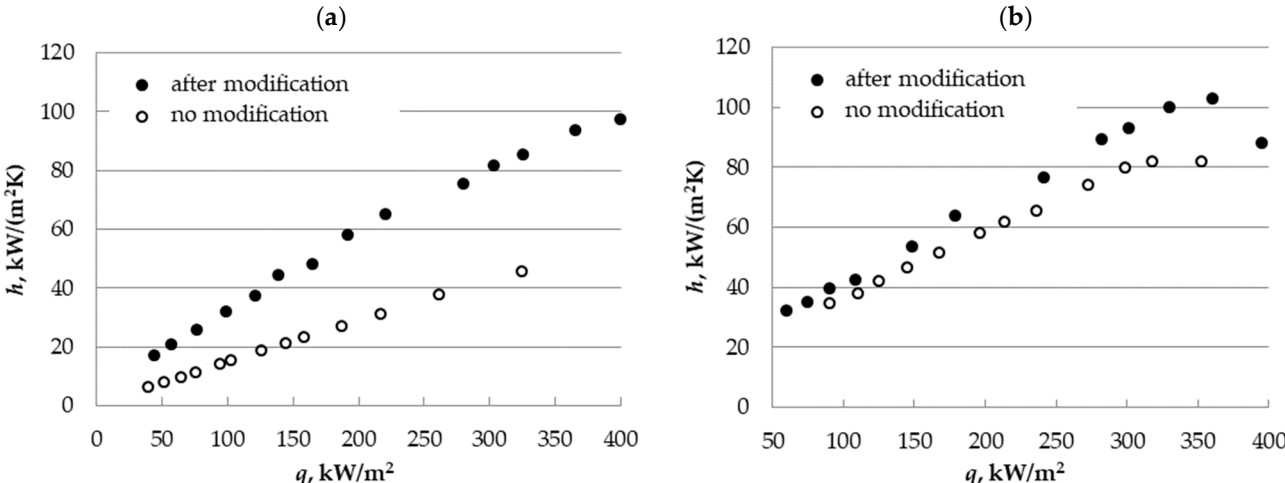

**Figure 8.** Effect of the modification on HTC at $G = 1260$ kg/(m$^2$s) in the T4 thermocouple section, (**a**) $p_r = 0.43$, (**b**) $p_r = 0.56$.

Figure 9 shows a comparison of the dependences of the convective HTC on the heat flux density for the non-modified and modified tubes for a maximum mass flow rate, at which the effect of convection is maximum. The data are in good agreement, indicating that there is no significant effect of the surface modification on the convective heat transfer compared to the non-modified tube. It can be concluded that the increased HTC in the nucleate boiling regimes, resulting from the modification, occurred due to the vaporization process. This conclusion is supported by the obtained boiling curves. Figure 10 shows an example of the boiling curves obtained before and after modification with the same inlet flow parameters. Comparing the values of $\Delta T_s = (T_w - T_s)$ at the constant value of the heat flux density $q \approx 250$ kW/m$^2$, it is clear that $\Delta T_s$ for the modified surface decreased by about 1.5 degrees, which indicated the intensification of boiling.

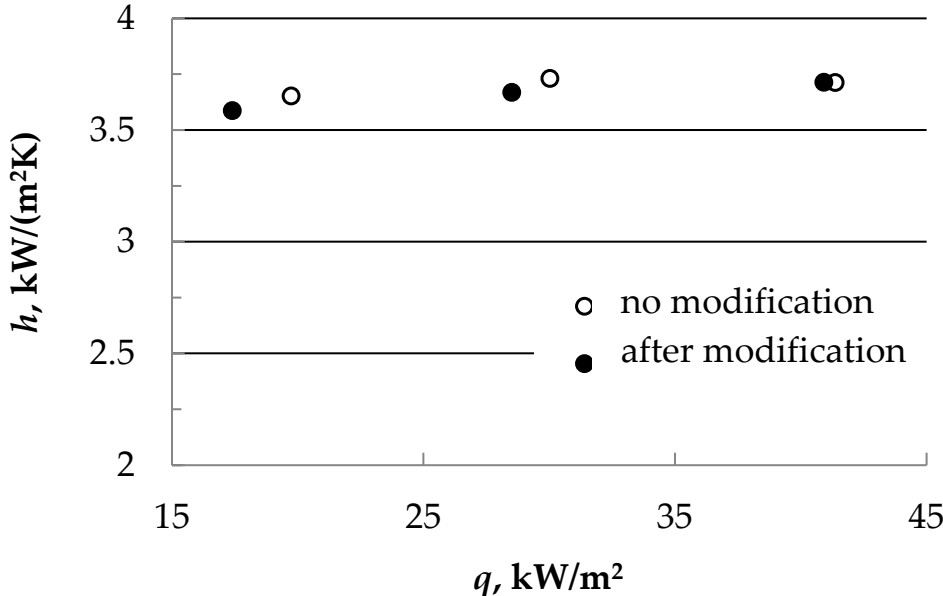

**Figure 9.** Convective HTC in the T4 thermocouple section versus the heat flux density at $p_r = 0.56$ and $G = 1260$ kg/(m$^2$s).

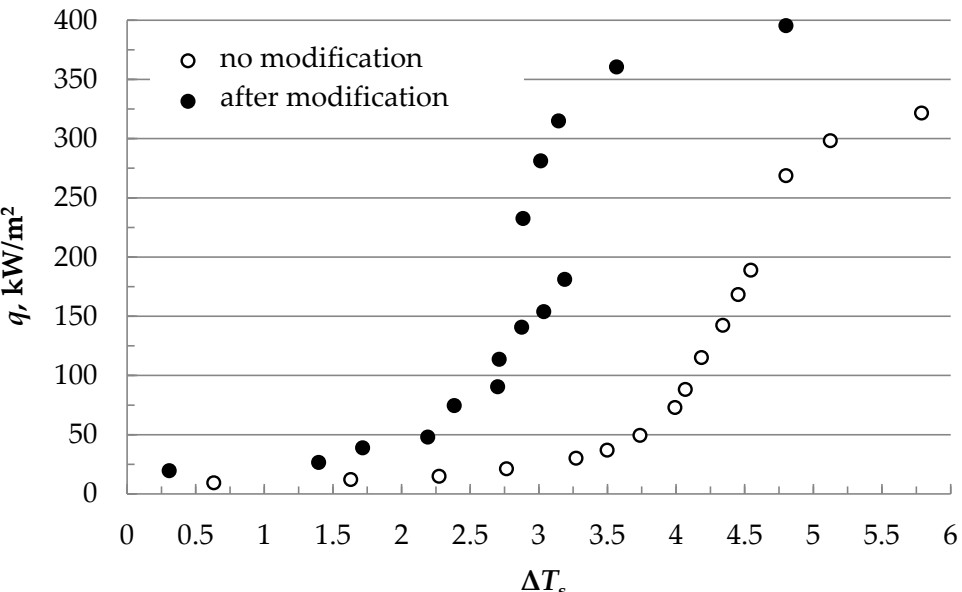

**Figure 10.** Boiling curves for T4 thermocouple section at $p_r = 0.43$ and $G = 800 \, \text{kg}/(\text{m}^2\text{s})$.

### 3.3. Effect of Intensification on the CHF

In each regime, the tube heat flux was gradually increased at a fixed mass flow rate and inlet temperature.

Depending on the flow parameters, the heat flux increase stopped either when departure nucleate boiling (DNB) or dryout occurred. The critical heat flux values were recorded when there was a sharp increase in the wall temperature at the outlet tube part in the T5 thermocouple section.

A comparative analysis of the CHF data obtained before and after the channel modification was performed. Figure 11 shows a comparison of the results for the CHF values dependent on the mass flow rate at different reduced pressures. The analysis showed that the modification to the inner wall caused an increase in the CHF for the range of mass flow rate $G > 800 \, \text{kg}/(\text{m}^2\text{s})$: the average increase was 22% at $p_r = 0.43$ (Figure 11a) and 7 % at $p_r = 0.56$ (Figure 11b).

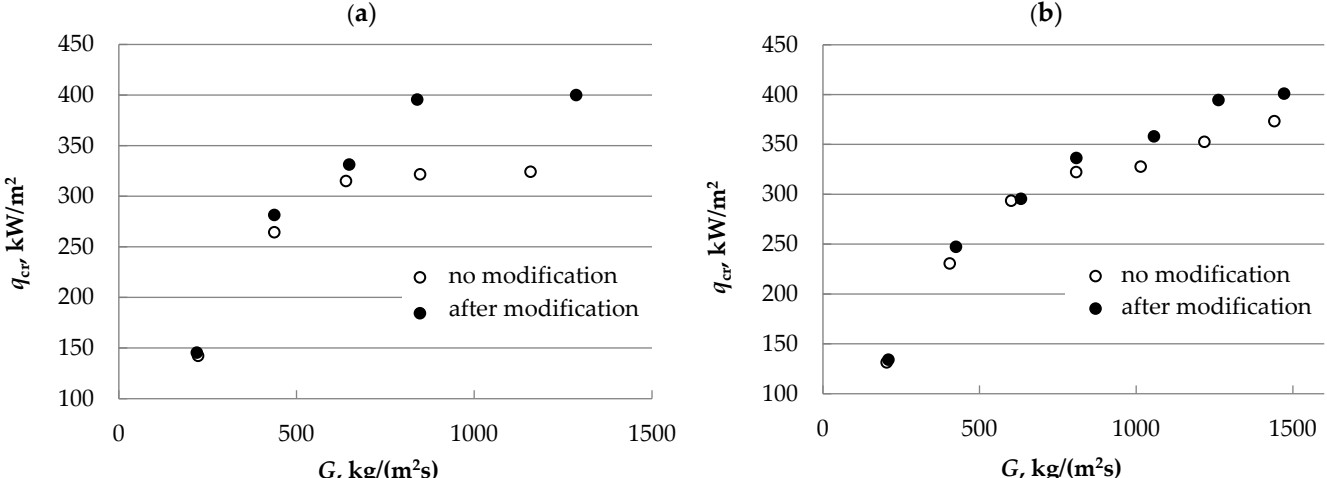

**Figure 11.** CHF data versus mass flow rate, (**a**) $p_r = 0.43$, (**b**) $p_r = 0.56$.

In the range $G < 800 \, \text{kg}/(\text{m}^2\text{s})$, no significant increases in the CHF were observed. As can be seen from the same CHF data versus vapor quality shown in Figure 12, no increases were observed because the heat transfer crisis occurs in the range of vapor quality $x > 0.7$ when the mass flow rate is decreasing. Dryout conditions also occur at this point.

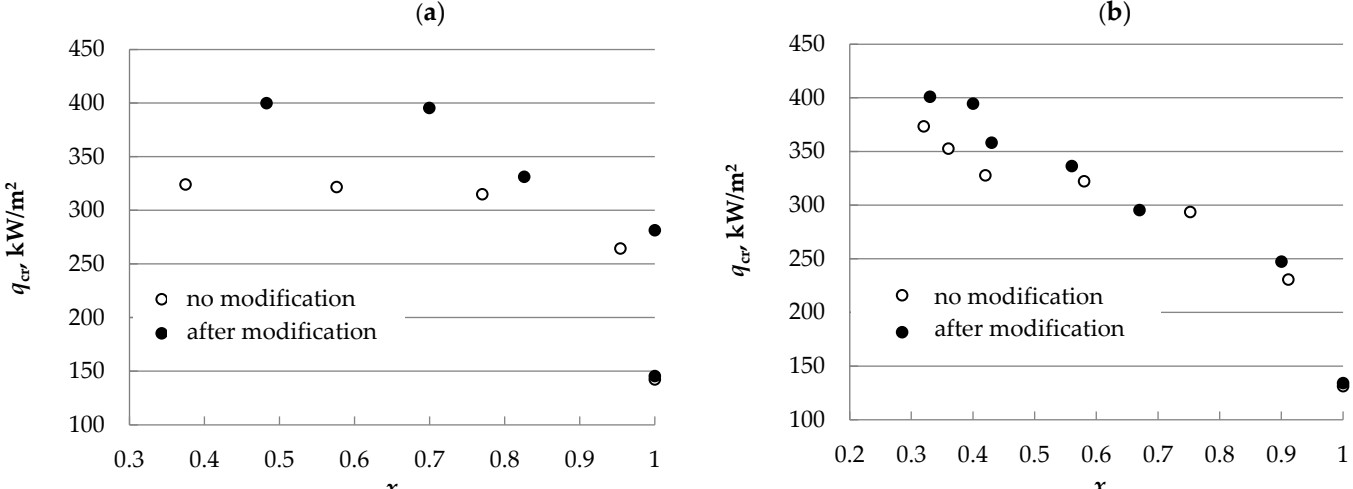

**Figure 12.** CHF data versus vapor quality, (**a**) $p_r$ = 0.43, (**b**) $p_r$ = 0.56.

Therefore, the greatest effect of the modification on the CHF was observed at the moderately reduced pressure $p_r$ = 0.43 in the range of the mass flow rate $G > 800$ kg/(m²s) where the average increase was 22%.

## 4. Conclusions

A method for modifying the inner surface of a mini-channel by treatment of the outer wall with a laser pulse has been developed.

We completed experimental studies on the effect of the modification to heat transfer during flow boiling of R-125 in the vertical channel with a diameter of 1.1 mm and length of 50 mm at two values of the reduced pressure: $p_r$ = 0.43 and $p_r$ = 0.56. The mass flow rate was varied in the range of $G = 200$–1200 kg/m²s. The maximum heat flux was limited by the CHF.

As a result of the modification, the heat transfer coefficient and the CHF increased. The greatest effect of the surface modification on heat transfer was observed at the reduced pressure $p_r$ = 0.43, where the increase in the HTC was up to 110% in the range of the mass flow rate $G$ = 1260 kg/(m²s). The maximum increase in the CHF was 22% at the reduced pressure $p_r$ = 0.43. It should be noted that the pressure increase led to a significant increase in the HTC by up to 100%. At the highly reduced pressure $p_r$ = 0.56, the effect of the surface modification on heat transfer decreased, where the HTC increased from 11% to 23% depending on the mass flow rate.

These research results showed that such modifications can be used to increase the intensity of heat transfer in various heat exchange devices, where low and moderate reduced pressures can also be used. Our experiments have shown that at moderately reduced pressures of $p_r \approx 0.4$, the values of the HTC can be significantly increased to those observed at high reduced pressures of $p_r \approx 0.6$ when using the presented method of inner surface modification in circular mini-channel.

**Author Contributions:** Writing-review and editing, A.V.B. and R.X.; investigation, A.V.B.; Conceptualization, A.V.D. and A.N.V.; supervision, A.V.D.; project administration, A.V.D.; funding acquisition, A.V.D.; software, N.E.S.; formal analysis, N.E.S. and P.J.; data curation, P.J.; methodology, A.N.V.; data curation, R.X. All authors have read and agreed to the published version of the manuscript.

**Funding:** This work was supported by RSF Grant 19-19-00410.

**Conflicts of Interest:** The authors declare no conflict of interest.

## Nomenclature

| | |
|---|---|
| $d$ | diameter, m |
| $G$ | mass flow rate, $kg/(m^2 s)$ |
| $p$ | pressure, Pa |
| $T$ | temperature, K |
| $x$ | vapor quality |
| $w$ | velocity, $m/c^2$ |
| Greek symbols | |
| $\alpha$ | heat transfer coefficient, $W/m^2 \cdot K$ |
| $\xi$ | hydraulic friction factor |
| $\rho$ | density, $kg/m^3$ |
| Subscripts | |
| l | liquid |
| g | gas |
| boil | boiling |
| con | convective |
| calc | calculated |
| exp | experimental |
| cr | critical |
| in | inlet |
| s | saturated |
| r | reduced |

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
