# Peer review of "Flow Boiling Heat Transfer Intensification Due to Inner Surface Modification in Circular Mini-Channel"

_water, doi:10.3390/w14244054_

Round 1
Reviewer 1 Report
The paper present an experimental study of the intensification of the boiling heat exchange by means of texturizing the inner surface of a circular milimetric-diameter circular channel, through which a flow of R125 freon is stablished. The research itself is moderately useful and could perhaps merit publication in the journal Water, but only after *very* mayor and important changes are introduced in the manuscript. My current recommendation is just to reject it.
1) The most important of the problems with the manuscript happens in the introduction. The authors introduce three figures in there that are *virtually direct copies* of figures published in other journals by researchers that are not coauthors of the present paper:
-Figure 1 of the present paper is a copy of Fig1a of ref[27]
-Figure 2 of the present paper is a copy of Fig.2 of ref[28]
-Figure 3 of the present paper is a copy of Figs.3c and 3d of ref[29]
(Besides, ref.28 itslef is incorrectly cited – the abbreviation of the journal should be Energy, not Int. Energy). I am not aware of any mention in the paper, or in any accompaning documentation, of holding the copyrights to use such figures and photographies. But even if the authors really hold such rights, it would be highly nonstandar to reproduce such figures that, in fact, contribute almost negligible to the reports of original results in this paper while at the same time give a very misleading view to the reader about who is the author of such material.
I can only emphasize that the editors should consider whether the submission of these figures constitutes unethical behaviour, or simple lack of expertise from the authors’ part. They should also ponder any possible precedent from the same authors. I feel that I cannot decide between these options with the reduced information and time available to me.
2) The authors use various acronyms and/or definitions of physical quantities that are never defined in the manuscript. For self-completeness of the paper, they should be defined, either in the text or in the final “nomenclature” section. Examples are “CHF” and “reduced pressure pr=p/pcr” in the abstract itself (that should be self-complete – besides, none of those two concepts seem even to be defined elsewhere in the manuscript). HFC22 and the rest of refrigerants in line 50, FC-72, etc.
3) Same for reduced pressure in line 175
4) In lines 89-90, the conditions for which pressure drop more than double should be given (for what dimensions of the pipe, tubulent flow or not, etc.)
5) In line 93, when the authors refer to “the results of the research” it is unclear what research the authors are talking about. They should clarify that sentence.
6) In line 123, the authors should specify the type of steel they use for their channel. They should also mention that the channel is of cylindrical section (what is at present only suspected from the title of the manuscript).
7) The authors should elaborate on how “The inner wall temperatures were calculated using a correction for the wall conductivity” (lines 129-130)
8) In section 2.2 (lines 136 and following) the authors mention the surface modification. However, they should provide more details on the type of laser employed, and on how they operate it (laser model, etc). It is difficult to understand, for instance, how they modify the inner channel surface by operating the laser “on the outer surface of the tube” (lines 137-138). When they mention the “optimal values of the pulse power”, they should especify numbers (and why they are considered to be optimal) (line 140). They mention a number of times the “technological current value” of the laser, in Ampere – what is the meaning of this electrical current? Why not state the watts, the seconds, and watts/square centimeters, of the light pulse itself?
9) The authors should give additional explanations of the origin of equations 2 and 3 (lines 198,199)
10) In section 3.1, the authors use the concept of a “crisis of heat transfer” (lines 224 and 225, and other). They should explain at a *quantitative* level what they mean.
11) In lines, e.g., 281 and 282, the authors mention the nucleate boiling regimes. They should brielfy explain the concept and, more important, how they identify that the experiment happen in that regime.
12) English of lines 28-29 should be improved.
13) Last but not least, the authors should use a broader range of fluids and channels in their studies, that seem to be too specific by now.
Author Response
Thanks for your review. We have taken into account your comments and made some corrections.
1) Your remark is correct. References to figures 1,2,3 have been added. Reference [28] has been edited.
2), 3) We tried to determine all physical quantities. pr is defined on line number 20.
4) This paragraph is about the conditions that were realized in the work [28].
5) Thank you for your comment. Reference [28] has been edited.
6) The type of steel “12X18H10T” has been edited. The title indicates a circular channel. Perhaps it was not a very good determination.
7) Here, the typical stationary task of the thermal conductivity of the wall is solved. In the opinion of the authors there is no need to give a well-known procedure of the solution here.
8) On this laser machine, power readings are given only in terms of the current that is supplied to the lamp. This parameter is technical. Of course, for a more detailed study of the process, it is necessary to know the exact amount of energy absorped for surface transformation, taking into account scattering, reflection, etc. For this purpose, we plan to carry out separate studies in the future. The phrase “optimal values of the pulse power” was unsuccessful and was replaced by “values of the pulse power”.
9) These formulas were taken from [33]. It is standard calculation method.
10) Thank you for your comment. The text has been edited.
11) The boiling regimes were determined from the wall temperature readings (Fig. 8) and from the boiling curves (Fig. 13).
13) Certainly, the work needs to be expanded. This requires more research in the future. At the moment, the results obtained are typical for the entire class of liquids commonly used in heat pumps and refrigeration units, as well as used for cooling microelectronics, and are self-sufficient. In the future, studies are planned not only on cylindrical channels, but also on model heat exchangers.
Reviewer 2 Report
The work is well written and provides good results, which are properly presented in the graphs. The proposed methods and techniques are interesting for readers who are working in related fields. However, the following points must be addressed before it is accepted for publication.
1-Significant results can be included in the abstract
2-The advantages of the surface modification must be clear
3-Special cases can be included in the results with comparison.
4-Conclusions must be re-written as a clear and summarized point.
5-Can the authors explain the pulse laser coast of this modification
6-related work can be included:
Key Influence Factors on Suspension Compressible Fluid through Pore Induced by SAW (Motion of Black Powder in Pipeline), Archive of Applied Mechanics, Published: 06 February 2021 https://doi.org/10.1007/s00419-020-01866-1
Author Response
Thanks for your review. We have taken into account your comments and made some corrections.
1) The abstract has been expanded with the phrase «The heat transfer coefficient increased up to 110%. The maximum increase in CHF was 22% at reduced pressure pr=0.43.»
2) 5) Such a laser system is a widely available technology. The surface treatment of a 50 mm long tube takes a minimum amount of time, which is a big advantage over other surface modification technologies.
3) The specifics of the studies performed are unique. To date, most surface modification studies have been performed on open surfaces. There are not so many works on surface modification inside the minichannel. So far, there are no other studies that can be compared with the results.
4) Thank you for your comment. The text has been edited
6) Thank you for your offer. The topic of your work is related to the modification of the fluid, not the surface. This is too far from the topic of our article.
Reviewer 3 Report
The authors presented an interesting and research-worthy topic, although I am not sure if "Water" is the right journal to present the research performed the field of heat transfer in a non-aqueous environment...
The authors should supplement the information on: (1) parameters of the R-125 medium in Table 1, (2) the calculation method for the hydraulic friction factor for not smooth walls (turbulent or laminar flow?).
In general, I have no major comments except for a few concerning the language, marked in the attached file. Also, I think the manuscript should be proofread by a native speaker.

Author Response
Thanks for your review. We have taken into account your comments and made some corrections.
1) Parameters of the R-125 have been addet to the table 1.
2) The calculation method for the hydraulic friction factor was for turbulent flow. Calculation method has been added to formula 1.
We have corrected the text thanks to your comments. Thanks a lot.
Reviewer 4 Report
This manuscript developed a method for modifying of inner surface of the mini-channel by treatment the outer wall with laser pulse. It can be published on Water with minor revision. But some of the irregularities were found and there were still some mistakes left in the manuscript. Following questions should be responded detailedly.
1. Please unify the format, such as K or ℃, and adjust the position and size of figures for better typography. Moreover, the expression of Figure in the text should be unified, such as “Fig.” or “Figure”. Furthermore, in line 127, “a cable diameter of 0.7”is missing unit. In Table 3, the unit may be lost, such as “Mass flow rate”, “Wall temperature”, “Inlet and outlet pressure sensors”, “Pressure drop sensor”. The expression of the range may be incorrected, such as “…in the range of mass flow rates G = 200 ÷ 1400 kg/(m2s)…” in Abstract, “…in the range G = 200 ÷ 1300 kg/m2s” in Line 177, “…in the current work (200 ÷ 1300 kg/m2s), …” in Line 216.
2. The results of this study should be state clearly in the abstract.
3. The method should be state clearly in the introduction, and how are the innovations of this study reflected?
4. On page 3, mentioned “The saturation temperature... in the range of 30-45 °C”, why the saturation temperature in Table 1 is -48.1 °C?
5. Figure 4 was obtained by what device? What is the pulse power P1-P14 value? What is the optimal pulse power value? Please check and modify.
6. On page 6, Figure 6b is mentioned, but where is the figure 6b? Please check and confirm.
7. On page 7, mentioned the maximum increase in pressure drop is not more than 40% in Figure 7. Please explain the “The greatest increase in pressure drop from 50% to 90%”.
8. Five cross-sections are selected in this paper, why only T4 is analyzed?
Author Response
Thanks for your review. We have taken into account your comments and made some corrections.
1) Thanks for your comments. We have tried to take them into account. The text has been edited.
2) The text has been edited.
3) Such a laser system is a widely available technology. The surface treatment of a 50 mm long tube takes a minimum amount of time, which is a big advantage over other surface modification technologies. To date, most surface modification studies have been performed on open surfaces. There are not so many works on surface modification inside the minichannel. So far, there are no other studies that can be compared with the results. At the moment, the results obtained are typical for the entire class of liquids commonly used in heat pumps and refrigeration units, as well as used for cooling microelectronics, and are self-sufficient. In the future, studies are planned not only on cylindrical channels, but also on model heat exchangers.
4) The temperature is given at one atmosphere.
5) On this laser machine, power readings are given only in terms of the current that is supplied to the lamp. This parameter is technical. Of course, for a more detailed study of the process, it is necessary to know the exact amount of energy absorped for surface transformation, taking into account scattering, reflection, etc. For this purpose, we plan to carry out separate studies in the future. The phrase “optimal values of the pulse power” was unsuccessful and was replaced by “values of the pulse power”.
6) The text has been edited.
7) The greatest increase in pressure drop from 50% to 90% is related to pressure drop in the tube in range of mass flow rate G > 2000 kg/m2s.
8) Thank you for your comment. We could show the results for all thermocouples, they are similar. However, we tried to select data on heat transfer for flow parameters that are most in demand in technology where the most developed boiling is observed. Thermocouples T1 and T2 are located at the entrance to the tube, where the flow is just beginning to be saturated with the gas phase. At the outlet, where the T5 thermocouple is located, the flow parameters can be close to the heat transfer crisis. Therefore, the T4 section was chosen. T3 showed similar results.
Round 2
Reviewer 1 Report
1) As I mentioned in my previous review, I do not consider acceptable thereproduction of the figures of other authors (or of pieces of them) in
the introduction when these figures do not contribute to the body of new
results presented in the paper (it would be different, e.g., if the paper
was a review, or if the paper was reanalyzing data shown in the figures of
those previous works – none of that applies to this paper). Therefore,
I believe that figures 1, 2 and 3 must be removed from the present paper.
References to them in the text must be either supressed or changed as follows:
-In line 79, substitute “Fig. 1 shows…” by “See Fig. 1a of [27] showing…”
-In line 90, substitute “(Fig. 2)” by “(see Fig. 2 of [28])”
-In line 102, substitute “(Fig. 3)” by “(see Figs. 3c and 3d of [29])”
In my professional opinion, this change is very easy to do,
but it is not “minor” because of its importance.
2) The authors provided reasonable answers to the rest of the comments
in my report
3) Therefore, I recommend rejection of the paper until the authors make the changes
I mention in point 1. This is, certainly, not a scientific-argument issue, but rather an
editorial practices aspect. Therefore, I leave in the hands of the publisher/editor to
decide whether my recommendation has to be followed or not. I do not need to revise
the manuscript again in further review rounds, because this aspect (whether these
figures are removed or not) could eventually be rapidly checked by the
editors/publishers.
Author Response
Thanks for your reasoned comments. We have removed figures 1,2 and 3.